# A Practical Yet Accurate Real-Time Statistical Analysis Library for Hydrologic Time-Series Big Data

Jun Sun [1], Feng Ye [1,*], Nadia Nedjah [2] , Ming Zhang [3] and Dong Xu [4]

1 School of Computer and Information, Hohai University, Nanjing 211100, China
2 Department of Electronics Engineering and Telecommunications of the Engineering Faculty, State University of Rio de Janeiro, Rua São Francisco Xavier 524, Marcanã, Rio de Janeiro 20550-013, Brazil
3 Water Resources Department of Jiangsu Province, Nanjing 210029, China
4 College of Water Conservancy & Hydropower Engineering, Hohai University, Nanjing 211100, China
* Correspondence: yefeng1022@hhu.edu.cn

**Abstract:** Using different statistical analysis methods to examine hydrologic time-series data is the basis of accurate hydrologic status analysis. With the wide application of the Internet of Things and sensor technologies, traditional statistical analysis methods are unable to meet the demand for real-time and accurate hydrologic data analysis. The existing mainstream big-data analysis platforms lack analysis methods oriented to hydrologic data. In this context, a real-time statistical analysis library based on the new generation of big data processing engine Flink, called HydroStreamingLib, was proposed and implemented. Furthermore, in order to prove the efficiency and handiness of the proposed library, a real-time statistical analysis system of hydrologic stream data was developed based on the concepts available in the proposed library. The results showed that HydroStreamingLib provides users with an efficient, real-time statistical verification method, thus extending the application capabilities of Flink Ecology in some specific fields.

**Keywords:** hydrologic information; statistical analysis; Flink; stream data; time series

## 1. Introduction

Hydrologic data are the core content for studying physical hydrologic processes, and simulating and predicting disasters, as well as monitoring water quantity and quality [1]. It is of great significance to analyze and master the temporal distribution of data on water level, velocity, etc. from rivers and lakes for water resource management and flood control. Accurate and real-time hydrologic forecasting also plays a vital role in the operation and maintenance of key infrastructures, such as dams and hydropower stations [2]. Statistical analysis is not only a common method for hydrologic time-series data analysis, but the basis for accurate hydrologic status analysis and decision. Through statistical analysis of hydrologic time-series data, the relevant characteristics of hydrologic time series, such as normality, stationarity, and trend, can be mastered [3]. With the change of global climate and the increase in extreme severe weather, the uncertainty of hydrologic information becomes more prominent and the demand for real-time performance of hydrologic models becomes more stringent. Therefore, it is necessary to conduct real-time statistical testing of hydrologic sequential data.

Hydrologic model testing is the process of evaluating the performance and accuracy of a hydrologic model. The objective of hydrologic model testing is to determine whether the model can accurately simulate the behavior of the hydrologic system it represents, and to identify areas where the model may need to be improved. This is achieved by comparing the outputs of the model with observed data from the hydrologic system, such as streamflow, rainfall, and other meteorological variables. The results of hydrologic model testing can be used to validate the model, to identify its limitations and strengths, and to make improvements to the model so that it can better reflect the behavior of the real-world

system. In order to validate the accuracy of a hydrologic model, it is often tested against observed hydrologic data. This is where the relationship between statistical tests and model tests comes into play. By performing statistical tests on the observed data, one can gain a better understanding of its properties and characteristics. This information can then be used to inform the development and selection of a suitable hydrologic model.

However, there are currently two key problems in real-time analysis of hydrologic time-series data. The first is the inadequate application of statistical tests and the second is the lack of a method for real-time statistical testing of large-scale hydrologic stream data [3]. When studying hydrologic time-series data, some researchers often ignore data parameter tests and unsafely assume that some hydrologic analysis and modeling methods meet certain characteristics, often using just a single test method to conduct subsequent analysis and decision [3]. However, as hydrologic data are easily affected by various uncertain factors—such as environmental factors, human factors, and equipment failure—they usually do not fully possess the prescribed characteristics, and the usage of a single statistical test method often yields inaccurate analysis results.

Research works based on general hydrologic data analysis mainly uses SPSS [4], MATLAB [5], and other tools to analyze historical data based on long time scales, such as a year, a season, or a month. With the increase in the number of sensors deployed in hydrologic stations and their sampling frequency, the scale of hydrologic time-series data has expanded rapidly. Hydrologic data have problems, such as outliers, data out of order, and repetition. Under the condition of high concurrency, traditional solutions are difficult to use for calculating these characteristics effectively [6]. Mainstream big-data computing engines, such as Apache Flink [7] and Apache Spark [8], can be effectively applied to large-scale data stream processing scenarios. Spark is a multi-language engine for executing data engineering on single-node machines or clusters. Flink is known as a third-generation stream computing framework, which can be used for stateful computation on unbounded and bounded data streams. Flink supports exactly-once semantics and the unique Checkpoint mechanism, providing high reliability and fault tolerance. However, Flink's standard and extended libraries only support a small quantity of statistical test methods, making it difficult for users to conduct comprehensive statistical analysis of a hydrologic temporal sequence of data in order to ensure the accuracy of a characteristic test [3].

Having a well-designed and well-executed data management system can help ensure the accuracy, reliability, and scalability of time-series data analysis. In this paper, we adopt a time-series database to manage large-scale hydrologic time-series data. A time-series database is a database designed specifically to handle timestamped data, also known as time-series data. Time-series data are a sequence of data points collected at regular intervals over a period of time. A time-series database has several features that make it well-suited for time-series data, including high write throughput, indexing and querying on time, efficient compression, and scalability.

Therefore, in order to meet the real-time requirements and the need for accurate hydrologic data analysis, we proposed a Flink-based hydrologic data analysis library, termed HydroStreamingLib, which includes 19 kinds of validation algorithm applicable to hydrologic data in several categories. Hydrologic practitioners can easily construct real-time statistical analysis systems through this library. Considering that most hydrologic practitioners are not familiar with Flink, we built a highly available real-time analysis statistical system based on HydroStreamingLib at the same time. The contribution of this work can be summarized as follows:

- We proposed and implemented HydroStreamingLib, a library for hydrologic time-series analysis based on Apache Flink. The library contains 19 different hydrologic time-series testing methods in four categories, which expands the ecology of Flink in the hydrologic field.
- Based on HydroStreamingLib, a hydrologic stream data verification system was constructed, which can be applied to the statistical testing of large-scale, high-velocity

hydrologic stream data and provide real-time visualization of test results. It realizes a complete solution from data collection, transmission, and analysis to persistence and visualization.

- We applied HydroStreamingLib to a real-world problem and evaluated the algorithms available in the proposed library to analyze different aspects. Compared with other general methods and tools, HydroStreamingLib achieved better results in real datasets.

The remaining contents of this paper are organized as follows: Section 2 introduces the research work related to this paper. In Section 3, the functions and implementation of HydroStreamingLib are introduced in detail. In Section 4, an example of hydrologic real-time analysis system based on HydroStreamingLib is described. In Section 5, data analysis and performance comparison experiments are carried out with the normality test method as an example. In Section 6, the proposed work is summarized and prospected.

## 2. Related Works

Statistical test methods are often used to analyze hydrologic time series. Different statistical analysis methods are used to analyze the normality, trend, and stationarity, as well as other characteristics of hydrologic data, to evaluate the status of hydrologic data and facilitate subsequent research on water resources.

In [9], Machiwal et al. present a study of the precipitation variation trend of 31 grid points in the arid coastal areas of India over 35 years. Eight trend tests, including Kendall's rank order correlation (KRC), Spearman's rank order correlation (SROC), Mann–Kendall (MK), four improved MK tests, and innovation trend analysis (ITA), were used to test the trend of precipitation data. Tosunoglu et al. [10] undertook a study of the trend of runoff parameters in Turkey's Coruh Basin and tested the maximum duration of annual runoff and annual maximum runoff, respectively, by using the modified MK test and ITA method; however, the results obtained from the MK test and ITA method were inconsistent with each other. The results demonstrated the effectiveness of the ITA method in determining drought and water resource management. Machiwal et al. [11] investigated the uniformity of precipitation records of four monsoon months and seasons at 16 stations in the Saraswati River Basin in Gujarat. Cluster analysis results showed that the data from four rain stations within four months had significant geographical differences and interannual precipitation dynamic differences. Gois et al. [12] tested the normality and homogeneity of rainfall time-series data in Rio de Janeiro through two parameter test methods and two non-parameter test methods, and evaluated the test methods.

For the above work, the data are only tested for a long time-span, and the amount of tested data is small, which cannot meet the premise of guaranteed fine-grained analysis of hydrologic data. With the development of big-data technology, large-scale hydrologic environmental data have become easy to obtain, but working out how to effectively analyze and visualize the massive dataset has become a big challenge.

Brömssen et al. [13] propose several different types of methods for visualizing large-scale environmental data and illustrated these methods by showing the variable trends related to acidification recovery in Swedish river data during 1988–2017, proving that the trend of large-scale environmental data can be comprehensively explained by a generalized additive model with a few specific graphs. The above work is based on historical fixed data. It is noteworthy to mention that there are only a few works regarding the inspection of stream data, so the real-time performance of inspection and analysis is not guaranteed.

In recent years, real-time data analysis systems have been applied in medical care, finance, smart cities, and other fields [14–16]. These works utilize big-data frameworks to build real-time data analysis systems, in order to ensure reliable and fast real-time information processing as data volume and complexity expand. However, the current mainstream big-data framework does not provide great support for the data analysis needs of the hydrology field.

RStudio [17] is an integrated development environment for R, a programming language for statistical computing and graphics. RStudio integrates the statistical analysis

library in R language to provide a variety of statistical analysis methods. However, as a stand-alone analysis platform, the performance of RStudio, as it will be shown, is far lower than that of HydroStreamingLib when dealing with statistical analysis of large-scale data.

SparkR [18] is a statistical analysis tool with Spark as the core, which provides a library of various statistical methods. It uses Spark's distributed computing engine to enable large-scale data analysis from the R shell. In contrast to SparkR, HydroStreamingLib uses Flink as its computing engine. As a new generation of big-data computing engine, Flink is more suitable for real-time statistical analysis of stream data, with higher computational efficiency [7].

Currently, with the increasing deployment of hydrologic sensor nodes, large-scale stream data need to be verified and computed in real time. Note that traditional schemes cannot guarantee the real-time performance and correctness of the verification under high load. Therefore, we proposed and implemented HydroStreamingLib and a sample system based on Flink. This study is different from existing related works, as it applied hydrologic statistical test methods to data streams and analyzed hydrologic data within a shorter time span.

## 3. The Proposed Library

This section mainly introduces the functions and implementation of HydroStreamingLib. It is based on the Flink framework and allows users to build a distributed computing platform for hydrologic time-series analysis by calling it. HydroStreamingLib can also be used for simultaneous, real-time stream data and large-scale historical hydrologic time-series analysis.

Data-processing based on Flink is mainly divided into three parts: data import (Source), algorithm component (Operator) and data export (Sink) [19]. HydroStreamingLib is an algorithmic component, and it supports a variety of data sources and data export forms. The location of HydroStreamingLib in Flink is shown in Figure 1. An example implementation of the system platform will be discussed in Section 4.

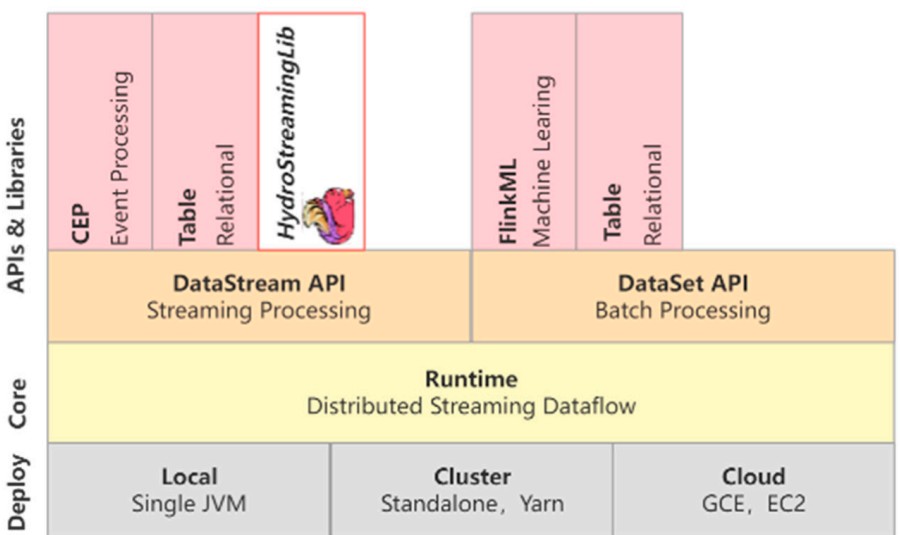

**Figure 1.** HydroStreamingLib in the Flink stack.

### 3.1. Statistical Test Methods

A total of 19 different hydrologic time-series analysis methods are implemented in HydroStreamingLib, including normality analysis, stationarity analysis, trend analysis and homogeneity analysis. The list of methods is shown in Table 1.

**Table 1.** Statistical methods.

| Characteristic | Methods |
|----------------|---------|
| Stationarity | Student's *t* Test |
| | Simple *t* Test |
| | Mann–Whitney Test |
| Normality | Kolmogorov-Smirnov Test |
| | Jarque–Bera Test |
| | Geary's Test |
| | Coefficient of Variation Test |
| Trend | Kendall Rank Order Correlation Test |
| | Adjacency Test |
| | Difference Sign Test |
| | Mann–Kendall Test |
| | Spearman's Rank Order Correlation Test |
| | Turning Point Test |
| | Inversions Test |
| Homogeneity | Bartlett's Test |
| | Bayesian Test |
| | Dunnett's Test |
| | Hartley's Test |
| | Von-Neumann's Test |

Normality analysis, stationarity analysis, trend analysis, and homogeneity analysis are all statistical techniques used to analyze time-series data. Normality analysis tests the assumption that the data are normally distributed. This is important in many statistical techniques that rely on this assumption.

Stationarity analysis is used to determine if time-series data are stationary, meaning that the mean, variance, and autocovariance are constant over time. Stationarity is an important assumption in many time-series models, including the ARIMA and ARMA models. Trend analysis is used to identify and analyze trends in the time-series data. This can involve fitting a trend line to the data or using statistical techniques, such as the moving average or exponential smoothing. Trend analysis is useful for understanding long-term patterns in the data and can be used for forecasting purposes. Homogeneity analysis is used to test if time-series data are homogeneous, meaning that they have consistent characteristics over time. This can be important in climate studies; for example, where changes in the data could indicate changes in the underlying climate.

In summary, these techniques can be used to analyze different aspects of time-series data and are useful for understanding the underlying patterns and trends in the data.

### 3.2. Data Importation and Preprocessing

HydroStreamingLib provides multiple forms of data access, including reading real-time data from external systems and message queues, or emulating historical data from a database. Hydrologic data from the sensor has strong real-time performance, high sampling frequency, etc. When the data are imported there may be some repetition, which can cause delays and unavoidable errors. Abnormal data will have a detrimental effect on the hydrologic time-series analysis and may cause characteristic judgment errors. Therefore, it is necessary to preprocess data to reduce the influence of abnormal data patterns.

The data preprocessing module of HydroStreamingLib provides additional exception handling capabilities, such as calling algorithms to remove duplicates, providing multiple outlier detection methods, and applying missing-value filling methods.

### 3.2.1. Data Duplication

The cause of data duplication is usually related to the communication handshake protocol between the sender and the receiver. Since sensor data are sampled at a certain

frequency, data duplication means that the data receiver receives multiple data with the same timestamp. If the data are not processed, the final analysis result will be affected to a certain extent.

The proposed HydroStreamingLib provides data duplication processing algorithms, designed for streaming data. Since the hydrologic sensor data are uniquely identified by a timestamp, when data from more than one sensor with the same timestamp appear in the preprocessing window, these data are judged as duplicate and thus removed. The algorithm maintains a timestamp array in the preprocessing window that bookkeeps the timestamps of all the data in the window. When new data arrive, their timestamp is compared with those held in the array. If the comparison results in a hit, the data are removed.

### 3.2.2. Data Anomaly

Hydrologic sensors that are exposed to harsh outdoor environments for extended periods of time may experience problems such as physical damage, electronic malfunctions, network transmission delay, and sensor degradation as a result of the adverse conditions. Meanwhile, hydrologic sensors are prone to interference and loss. So, abnormal data are inevitable in daily work. These abnormal data often contain values that are impossible to occur in practice. These outliers greatly affect the accuracy of subsequent statistical analysis, making anomaly removal one of the key steps for accurate and efficient data preprocessing.

In hydrological time series, outliers usually take the form of point anomalies; that is, the data at a certain timepoint differ significantly from the data collected during the period before and/or after it. At present, there are many works on anomaly detection in time series, most of which are based on similarity measurement and deep learning [20,21]. However, the high complexity of these approaches makes them unsuitable for high-speed, real-time streaming data [22]. Therefore, HydroStreamingLib provides high-performance algorithms based on statistical analysis for real-time anomaly detection, including ESD [23], BoxPlot [24], Hbos [25], and the Tukey test [26].

### 3.2.3. Missing Data

In the case of hydrologic sensors, there are two main reasons for missing data. The first is sensor failure or damage, which leads to a period of time during which the hydrological indicators record that data are not received. The other is data loss caused by network congestion or a receiver that does not receive data during the transmission. The statistical analysis of hydrological sensor data is very dependent on the continuity regarding the integrity of the received data. Therefore, if the data are missing, data in the studied time range will show a certain feature fully and correctly, thus affecting the final analysis of results.

In addition to the impact with regards to missing sensor data, if the received data also contains outliers, their deletion during the process of anomaly removal will further increase the proportion of missing data. In order to solve both problems, the approach proposed in this paper deals with different situations. When the missing data in the current window exceed a certain proportion, the data in that period is considered to lose statistical significance and the window is discarded altogether. Otherwise, if this proportion of missing data is not significant, the missing values will be obtained via a linear interpolation. Assume that there are $n$ data in the window, and there are $i$ consecutive missing values. So let $M_i$ ($2 \leq i \leq n-1$) be a window data value, wherein $M_1$ is the previous normal value of the missing data in this group, and $M_n$ is the last normal value of the missing data in this group. Then, we yield $M_i$ using Equation (1):

$$M_i = \frac{(i-1) \times M_n + (n-i) \times M_1}{n-1} \tag{1}$$

### 3.3. Distributed Implementation

This section takes the Kolmogorov–Smirnov Test as an example to discuss the distributed implementation of the algorithm. Firstly, it receives data continuously from the

external system, and the data contain multiple features (timestamp, sensor ID, measurement value, redundant information). Time and water level are registered according to the timestamp. Secondly, map operation is performed to extract information and package it into classes that can be recognized by Java. KeyBy operation is then performed to divide data according to sensor ID. The time window is divided and the data in the time window are preprocessed to remove outliers and redundant values. Then K–S test calculation is performed on the data in the window. The window size provides four timescales of 1 h, 1 day, 7 days, and 30 days. Finally, reduction operations are performed to integrate the calculation results and output them to the external system.

### 3.4. Characteristic Discrimination

HydroStreamingLib provides a variety of commonly used probability distribution tables for calculating p-values under different degrees of freedom to determine the characteristics of data flows. HydroStreamingLib returns a test value or *p*-value and provides the ability to check for normality, stationarity, tendency, homogeneity, etc. In the method of characteristic discrimination, the minority is subordinate to the majority while some factors of weak test methods are considered; such weak test methods are given low weight.

### 4. Hydrologic Real-Time Analysis System

Considering that most hydrologic practitioners are not familiar with big-data frameworks, it may be difficult for them to construct an integrated distributed real-time analysis system. Therefore, in this paper we provide an all-in-one solution for hydrologic practitioners who have no experience with big-data frameworks. Based on HydroStreamingLib, we perform a hydrologic real-time statistical analysis system from data acquisition to visual display. The basic process of the system is shown in Figure 2. Hydrologic data are collected by hydrologic sensors in rivers, and the hydrologic time-series data are transmitted to the Flink cluster for processing through the message queue Kafka. Flink receives and analyzes the data of different topics. In the Flink cluster, the data flow of each station is divided into sub-streams according to the station's identity. Meanwhile, the hydrologic time-series data of different sub-streams are aggregated according to the four time dimensions of hour, day, week and month. HydroStreamingLib is called to calculate the statistical value and identify the properties of the data in each time window. Finally, the test results are inputted into the message queue Kafka, and the test results of the real-time data displayed by the platform are verified. At the same time, the data are stored in the time-series database IoTDB for historical data querying. The system runs on a distributed cluster, and Flink allows work for a given computation, insert, or query to be divided into smaller sub-tasks that can be run concurrently on different nodes to reduce the overall processing time and improve the performance of the solution [27].

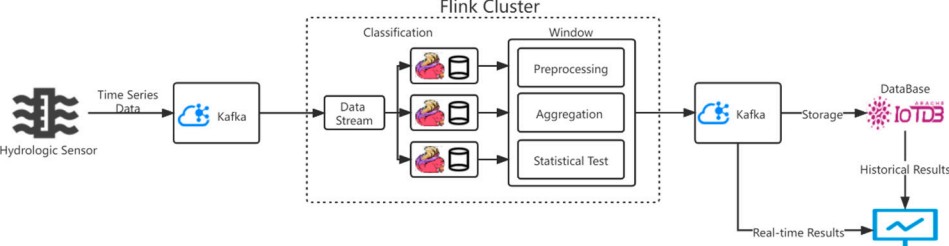

**Figure 2.** System architecture.

### 4.1. Data Aggregation Based on Time Window

In streaming data processing, low latency and resource requirements are in eternal conflict. On one hand, blindly pursuing the performance of streaming data processing needs expensive computing resources to support it. On the other hand, if computing resources are limited, the real-time performance of streaming data processing is difficult to guarantee. Therefore, a solution for how to improve the performance of streaming data

processing while keeping the computing resources unchanged is a hot issue in streaming computing research. Since hydrological time series essentially reflects the trend of a certain hydrological value changing over time, the time nature of the data should not be lost when the hydrological sensor data are processed, so as to ensure that the processed data can keep the various characteristics of the original time series in time. Using the aggregation idea and the window operator of Flink, we proceed with the data aggregation method based on time window. Its main steps of the method are as follows:

**Step 1.** A rolling time window $W_0$ is established for the hydrologic data stream $S_0 = (T_0, V_0)$ flowing into Flink, and the window size is set to 1 h (the window size can be determined according to the actual situation, and the default value in this paper is 1 h), where $T_0$ is the sampling timestamp of the sensor, and $V_0$ is the sampled hydrologic value.

**Step 2.** The average value over an hour of hydrological data is calculated in $W_0$, which is used to measure the centralized location of window data. Since the data have undergone a series of preprocessing prior operations, the mean value has good robustness at this time.

**Step 3.** After the data stream $S_0$ is calculated by $W_0$, the output data stream $S_1 = (T_1, V_1)$, $T_1$ is the timestamp with an interval of one hour, and $V_1$ is the mean sequence of hydrologic data every hour. At this point, a new scroll window $W_1$ is created with a window size of 24 h.

**Step 4.** The average of each day's hydrologic data is calculated in $W_1$. After the data stream $S_1$ is calculated by $W_1$, its output data stream $S_2 = (T_2, V_2)$, $T_2$ is a timestamp with an interval of one day, and $V_2$ is the mean series of daily hydrological data.

**Step 5.** The data stream $S_2$ contains the aggregated mean value of daily hydrological data, which can meet the statistical analysis in the time range of weeks, ten days, and months commonly used in the field of hydrology. If necessary, a new time window can be aggregated to $S_2$ so that it can continue to be created in a higher time dimension.

After aggregating hydrologic time-series data, some short-term water level change information will be lost. However, for statistical analysis with higher time dimensions, such as a week, ten days, or a month as the time range, the water level change information during an hour is not important and will not seriously affect the analysis results. Therefore, this redundant information can be removed to improve the computational efficiency.

### 4.2. Message Queue

The system uses a distributed message queue, Apache Kafka [28], as the data inflow and outflow pipeline. Kafka is a distributed message queue based on publish-and-subscribe mode, which is mainly applied in the field of real-time processing of big data. Kafka distinguishes between producers and consumers. Producers can publish messages to topics, which are then stored on a set of servers called brokers, from which consumers can subscribe to one or more topics and use the subscribed messages by extracting data from the brokers.

Kafka provides data caching, peak elimination, decoupling, and asynchronous communication. It is mainly used to solve the problem of mismatch between upstream and downstream data speeds and enables multiple applications to use the same data. Kafka implements its fault tolerance mechanism based on a distributed architecture. The Leader node in Kafka Broker synchronizes data to the Follower node. In the event of a Leader node failure, the downstream API interface can still receive data. Kafka is used as a data pipeline in this system.

### 4.3. Processing Module

The system uses Flink as a stream data processing tool. Flink [7] is a framework and distributed processing engine for stateful computations over unbounded and bounded data streams. Compared with other streaming data processing engines, Flink supports event-time semantics. When the event arrives at the stream data processing engine, Flink allows the actual event timestamp to be extracted and assigned to the corresponding window [7]. These features of Flink ensure the high efficiency and accuracy of real-time inspection of distributed stream data. The processing process of Flink is shown in Figure 3.

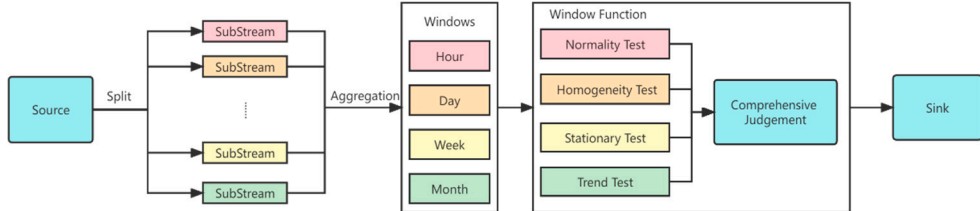

**Figure 3.** Flink processing flow.

The Flink data source, as the consumer of Kafka, reads the stream data from Kafka. After obtaining the data, it divides the data according to the identity of the hydrologic station and calls the calculation and discrimination module in HydroStreamingLib to calculate the statistical test results and judge the characteristic. One of the basic calculation discrimination modules is shown in Figure 4. Finally, the calculated results are put into the message queue as Kafka's producer for subsequent data storage and real-time display.

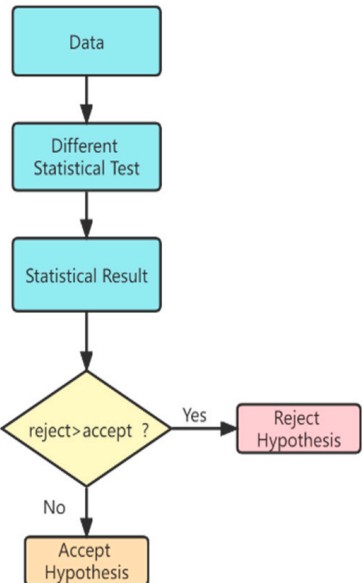

**Figure 4.** Discrimination module.

### 4.4. Data Storage and Real-Time Display

Apache IoTDB [29], a time-series database, is used in the system to store historical data and inspection results. IoTDB uses timestamps as the unique identification of data and provides a new tree file format called TsFile (time-series File). As the main storage format of IoTDB, TsFile can optimize the organization of time-series data, reduce the storage size and improve query performance. Hydrologic stream data sources and test results are time-series data, which can be effectively managed on IoTDB for subsequent query and visualization.

The system uses Vue and SpringBoot frameworks to implement a visual platform. Vue is a set of progressive frameworks for building user interfaces for building platform front ends. SpringBoot is an open-source application framework on the Java platform, which is used to build the backend of the platform. SpringBoot listens to the data in the topic corresponding to Kafka inspection results and transfers the data to Vue in real time. The platform provides real-time data visualization, historical data query, and display.

### 4.5. Platform Hardware Requirements and Availability

The hardware requirements, price, and availability of a hydrologic statistical analysis platform based on Flink would depend on several factors, such as the scale of the data, the required processing power, the desired level of performance, and the available infrastructure.

In terms of hardware requirements, a Flink-based hydrologic statistical analysis platform would typically require a cluster of machines with sufficient memory, CPU, and storage capacity to handle the data and processing workload. The exact hardware specifications would depend on the specific requirements of the application, such as the size and complexity of the data, the desired performance, and the budget.

The price of the application platform would also vary depending on several factors, such as the hardware requirements, the cost of the software licenses, and any additional services or support needed. It is worth noting that Flink is open-source software, which can help reduce the cost of the software license.

In terms of availability, a Flink-based hydrologic statistical analysis platform can be set up and deployed on-site or in a cloud environment, depending on the specific requirements and infrastructure of the organization. Some popular cloud providers, such as Amazon Web Services (AWS), Microsoft Azure, Alibaba Cloud, and Google Cloud Platform (GCP), offer Flink as a managed service, which can simplify the deployment and management of the platform. Hosting applications in the cloud platform improves availability and allows for timely error correction and maintenance of the system when there are different application requirements.

## 5. Experiment

We conducted experiments on the software package and the proposed system, and this section mainly expounds the experimental results. First, the dataset and experimental environment are introduced. Then, we discuss how we compared the computational efficiency of HydroStreamingLib with that of commonly used statistical test algorithm packages. We then describe how we used real datasets to simulate data flow to evaluate the performance of the system based on HydroStreamingLib. After that, we describe how we evaluated the performance of the system under higher loads.

### 5.1. Dataset and Experimental Environment

In this work, the water level data of 69 stations in the Chu River basin from 2016 to 2017 were collected and acquired. The dataset contained more than 20 million data samples, including 69 different sensors; each sensor was numbered using an 8-digit number format such as "12,910,280". The sampling interval of this dataset was 5 min, and its format is shown in Table 2 after redundant information was removed. Redundant information includes the name of the collection device, the identity of the collection person, and the description.

**Table 2.** Data formats.

| Timestamp | Station ID | Water Level/m |
|---|---|---|
| 2015-01-02 10:00:00 | 12910540 | 58.490 |
| 2015-04-23 13:55:00 | 62916400 | 5.480 |
| 2015-06-28 16:45:00 | 60403100 | 7.490 |

In this work, the experimental environment was set up on an Alibaba cloud cluster composed of four nodes. The hardware environment of each node was Intel(R) Xeon(R) Platinum 8269@2.50GHz CPU, 8 GB RAM and 50 GB HDD, and the system instance was 64-bit CentOS 7.6. The software environment was Apache Kafka 2.8.0, Apache Flink 1.13.1, Apache IoTDB 0.12.2, Apache Spark 3.1.1, and Prometheus 2.28.0.

### 5.2. Statistical Anlysis of Water Level in Chuhe River

In this section, the water level time series collected by station #12910280 on the Chuhe River during the second semester of 2016 was used as the dataset, which was simulated into flow data and transmitted to the Flink platform. The proposed HydroStreamingLib was used to analyze the time series for a 1-month time span. To demonstrate the results

of our statistical analysis, we selected three different statistical tests in each category as examples. This part of the experiment was mainly used to prove the application of our proposed HydroStreamingLib in practical application scenarios.

Table 3 shows the normality test results of the water level time series. It can be confirmed that the water level of all months showed obvious normality, except for November. This indicates that during the second semester of year 2016, the water level time series, sampled monthly, generally conforms to normal distribution.

**Table 3.** Result of normality test.

| Month | K–S Test | J–B Test | Geary's Test | Result |
|---|---|---|---|---|
| 2016.7 | **0.1236** | **2.4358** | **0.963** | ✓ |
| 2016.8 | **0.2054** | **2.4079** | 1.1314 | ✓ |
| 2016.9 | **0.1899** | **3.3242** | 1.135 | ✓ |
| 2016.10 | **0.1615** | **0.0902** | 0.8766 | ✓ |
| 2016.11 | 0.3099 | 714.6828 | 0.5949 | ✗ |
| 2016.12 | **0.2172** | **3.5783** | 1.1199 | ✓ |
| Threshold | 0.24 | 5.991 | 1 | — |

Table 4 shows the results of the stationarity test, which indicate that the water level data during the months of September, November, and December were relatively stable. The water level in November and December hardly fluctuated. So, the statistics of the two sub-series are almost the same, showing a strong stationarity. Although the water level data in September show a relatively clear change, the data series first shows a decreasing trend, and then a rising one. This indicates that there is little difference in the mean and variance between the two sub-series, so it also shows a certain stationarity.

**Table 4.** Results of stationarity test.

| Month | Student $t$ Test | | | Simple $t$ Test | Mann–Whitney Test | Result |
|---|---|---|---|---|---|---|
| | Subsequence 1 | Subsequence 2 | Subsequence 3 | | | |
| 2016.7 | **1.6793** | **0.778** | −2.4602 | **0.1524** | 0.0437 | ✗ |
| 2016.8 | 2.9517 | **0.7566** | −3.7088 | 0.0 | 0.0 | ✗ |
| 2016.9 | **1.6185** | −3.7532 | 2.1351 | **0.8519** | **0.7934** | ✓ |
| 2016.10 | −3.0603 | **0.0754** | 2.9856 | 0.0 | 0.0 | ✗ |
| 2016.11 | **0.2273** | **−0.0052** | **0.239** | 0.0121 | **0.9279** | ✓ |
| 2016.12 | **1.637** | −2.366 | **0.729** | **0.4429** | **0.6934** | ✓ |
| Threshold | 1.833 | 1.833 | 1.833 | 0.05 | 0.05 | — |

The homogeneity test results, presented in Table 5, show that—except for October—the time series regarding water levels for the remaining months exhibits uniformity. This proves that the water level data received from the sensors during the second semester of 2016 well reflect the natural changes of the Chuhe River itself, rather than unnatural factors, such as location changes and equipment failures.

**Table 5.** Results of homogeneity test.

| Month | Dunnett's Test | | | Bayesian Test | Von-Neumann's Test | Result |
|---|---|---|---|---|---|---|
| | Subsequence 1 | Subsequence 2 | Subsequence 3 | | | |
| 2016.7 | **0.3537** | **0.2006** | **1.6949** | 2.6918 | **2.0144** | ✓ |
| 2016.8 | 2.1877 | 2.7654 | 3.3344 | **1.1825** | **2.0237** | ✓ |
| 2016.9 | **1.5643** | **1.9273** | **1.4739** | 4.0711 | **2.0716** | ✓ |
| 2016.10 | 11.1853 | 13.6305 | 22.6544 | 13.3524 | 0.0456 | ✗ |
| 2016.11 | **2.1327** | **0.2324** | **1.5253** | **2.0354** | 0.7863 | ✓ |
| 2016.12 | 2.7363 | 2.5679 | 2.6607 | **1.8434** | **1.9006** | ✓ |
| Threshold | 2.15 | 2.15 | 2.15 | 2.42 | 2 | — |

Table 6 shows the trend test results of the water level time series. Unlike the other three-feature testing algorithms, the null hypothesis of the trend testing algorithm was "the time series has no trend". The results in the table show that the water level data during the months of July, August, and October exhibited an obvious trend, and there was a significant rise during October. The remaining months show neither a rising nor a decreasing trend. *NaN* in Table 6 indicates that the statistical characteristics of the time period could not be calculated.

**Table 6.** Results of trend test.

| Month | SROC Test | KRC Test | Mann–Kendall Test | Result |
|---|---|---|---|---|
| 2016.7 | −2.6142 | −2.9437 | 0.0034 | √ |
| 2016.8 | −18.6319 | −7.0472 | 0.0 | √ |
| 2016.9 | **−0.4335** | −0.4103 | 0.6947 | × |
| 2016.10 | *NaN* | 7.7608 | 0.0 | √ |
| 2016.11 | **−1.3136** | −1.6592 | 0.1007 | × |
| 2016.12 | **−0.8942** | −2.0517 | **0.0655** | × |
| Threshold | 2.048 | 1.96 | 0.05 | — |

*5.3. Contrast Experiment*

In this experiment, the HydroStreamingLib proposed in this paper, RStudio, and SparkR were used for statistical analysis of different scales of Chu River water level data. This part of the experiment was mainly used to compare the computational performance with the existing big-data statistical analysis framework.

We exploited the K–S test algorithm as the verification algorithm of the experiment. Since RStudio only supports stand-alone use, we deployed it on a single server, while HydroStreamingLib and Spark were deployed on different numbers of nodes for experiments.

The experimental results obtained after multiple experiments are shown in Table 7. Experimental results showed that HydroStreamingLib and SparkR were significantly more efficient in computing and processing data than RStudio in the standalone deployment. When the data size was increased to 1024 MB, the calculation time of the validation algorithm applied to RStudio was more than 1 h. SparkR with a 4-node computing cluster took 121 s, while HydroStreamingLib took only 65 s. Experiments showed that when processing data of the same size, HydroStreamingLib was computationally more efficient than SparkR and RStudio. Apache Flink and SparkR are both big-data processing frameworks, while RStudio is a data analysis platform. Flink is designed to provide low-latency data processing, making it well-suited for real-time data analytics applications. SparkR and RStudio, by contrast, are not optimized for low-latency data processing and may not perform as well for real-time data analytics. Flink provides in-memory processing, which enables fast and efficient data processing. SparkR also supports in-memory processing, but RStudio does not.

**Table 7.** Calculation time of K–S test algorithm on HydroStreamingLib, SparkR, and RStudio.

| Data Amount (MB) | HydroStreamingLib Run Time (s) | | | | SparkR Run Time (s) | | | | RStudio Run Time (s) |
|---|---|---|---|---|---|---|---|---|---|
| | 1 Node | 2 Nodes | 3 Nodes | 4 Nodes | 1 Node | 2 Nodes | 3 Nodes | 4 Nodes | |
| 32 | **7** | **5** | **3** | **2** | 19 | 16 | 13 | 7 | 179 |
| 128 | **29** | **18** | **15** | **12** | 62 | 35 | 27 | 21 | 711 |
| 512 | **109** | **56** | **37** | **31** | 249 | 131 | 108 | 79 | >3600 |
| 1024 | **209** | **105** | **77** | **65** | 432 | 246 | 163 | 121 | >3600 |

*5.4. Parallel Performance Experiment*

Performance parallelism is mainly measured using three classic metrics, namely Speedup, Sizeup, and Scaleup. In this experiment, three index results of the proposed

system and SparkR were compared under different system configurations, varying the numbers of nodes and different scales of data to evaluate the system parallel performance. This part of the experiment mainly compared the parallel performance of HydroStreamingLib with existing statistical analysis frameworks. The higher the parallelism, the more scalable the system.

5.4.1. Speedup Ratio

The speedup metric is used to measure the ratio of time spent in a single node and $m$ nodes under the same scale dataset. If the speedup ratio increases linearly with the increase in the number of nodes $m$, then we can conclude that the system can reduce the computation time linearly by increasing the number of nodes. The speedup ratio is defined as in Equation (2):

$$Speedup = t_1/t_m \tag{2}$$

Wherein $t_1$ represents the execution time using a single node and $t_m$ that when using $m$ nodes.

The speedup ratios of the system based on the proposed library and that based on SparkR are shown in Figure 5. It was easy to conclude that, with the increase in the number of nodes, the speedup ratios of the system and SparkR generally yielded a linear increase. When the number of nodes increased to four, the speedup ratio fluctuated around three. This shows that the computation time could be reduced approximately linearly by increasing the number of nodes.

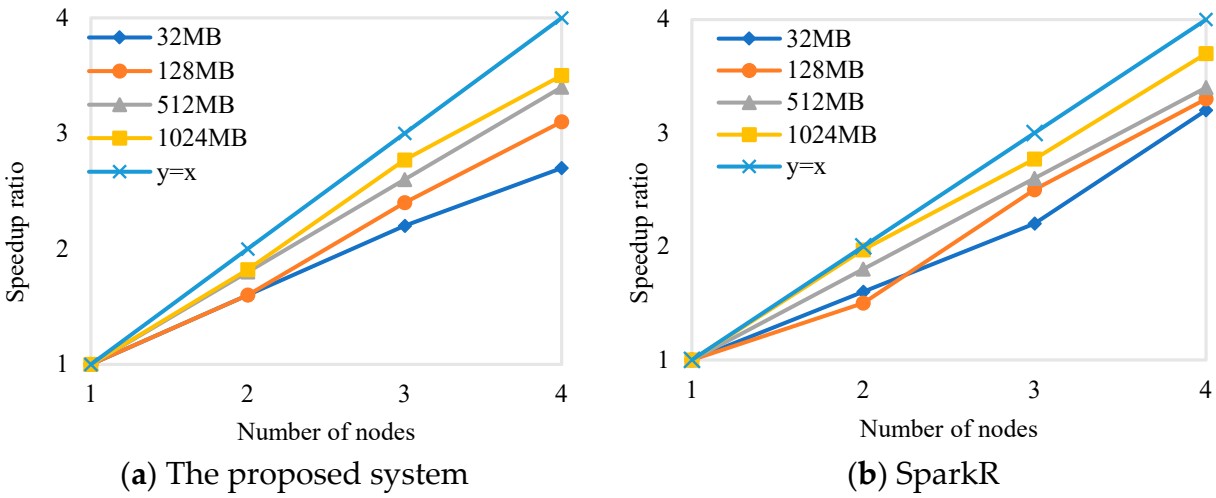

(**a**) The proposed system      (**b**) SparkR

**Figure 5.** Comparison with regards to speedup.

5.4.2. Sizeup Ratio

The sizeup metric measures the increase in computation time of the node when the number of nodes remains the same while the amount of data increases exponentially. The sizeup ratio is defined as in Equation (3):

$$Sizeup = t_m/t_1 \tag{3}$$

Wherein $t_m$ is the time taken to compute $m$ units of data on $n$ nodes and $t_1$ is the time taken to compute one unit of data on $n$ nodes.

Figure 6 shows the sizeup test results for the system based on the proposed library and that based on SparkR. We concluded that, as the amount of data increased exponentially, the increase ratio of computation time of the compared implementations was very close. When the amount of data increased from 32 MB to 1024 MB, the sizeup ratio fluctuated around 15.

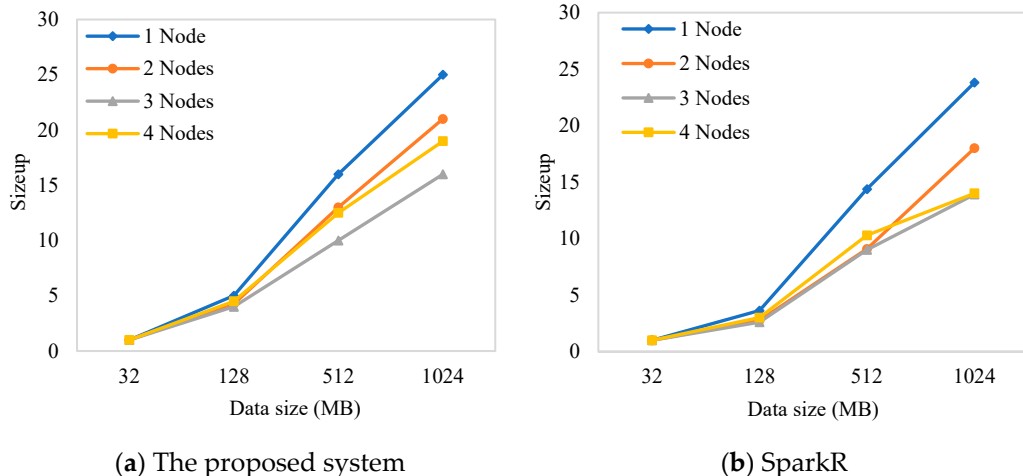

(**a**) The proposed system

(**b**) SparkR

**Figure 6.** Comparison with regards to sizeup.

5.4.3. Scaleup Ratio

The scaleup metric measures the ratio of the computation time of *m* units of data on *m* nodes to the computation time of a single data unit on a single node. With the increase in data volume, if the scalability index is lower than 1.0 but nearing 1.0, then one can conclude that the distributed system has good adaptability to the amount of data. The scaleup ratio is defined as in Equation (4):

$$Scaleup = t_1/t_m \tag{4}$$

Wherein $t_m$ is the time taken to compute *m* units of data on *m* nodes and $t_1$ is the time taken to compute one unit of data on a single node.

The obtained scaleup ratios for the compared systems are shown in Figure 7. It can be noted that under different data amounts, with the increase in the number of nodes, the scaleup ratios of the system based on the proposed library and that based on SparkR were kept between 0.8 and 1.0, which indicated that both systems exhibited good scalability and could provide support for data of different scales.

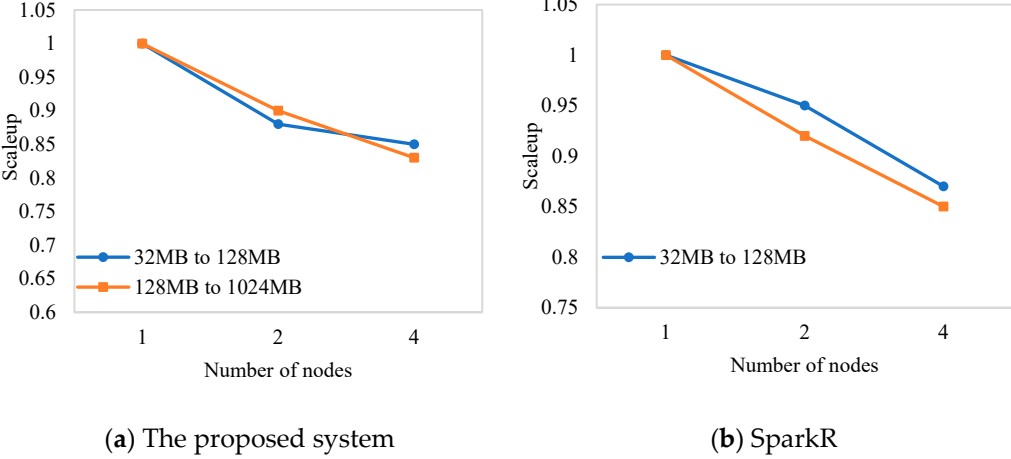

(**a**) The proposed system

(**b**) SparkR

**Figure 7.** Comparison with regards to scaleup.

In summary, compared with RStudio and SparkR, the system based on the proposed library had higher computational efficiency for statistical analysis of hydrological sensor data, yet it could make timely and accurate statistical analyses of more data in the same amount of time. In terms of parallel performance, the system based on the proposed library had similar performance to SparkR in terms of speedup, sizeup, and scaleup, which showed that the system based on the proposed library could process large-scale data by adding nodes, and thus had good parallel performance. The main reason for this

was that the combination of a big-data distributed computing framework and statistical analysis methods could fully schedule computing resources and effectively deal with high concurrency application scenarios.

### 5.5. Evaluating the System on a Real Dataset

Based on the Chu River dataset, we evaluated the real-time analysis system multi-dimensionally and monitored the key nodes of the system, such as Flink and Kafka, through Prometheus. This part of the experiment was mainly used to verify the resources consumed by the proposed system in real scenarios.

Due to the slow transmission rate of the original Chu River dataset, the transmission rate of the source data was increased to 500 ms/piece in the experiment. The CPU usage of each node is shown in Figure 8.

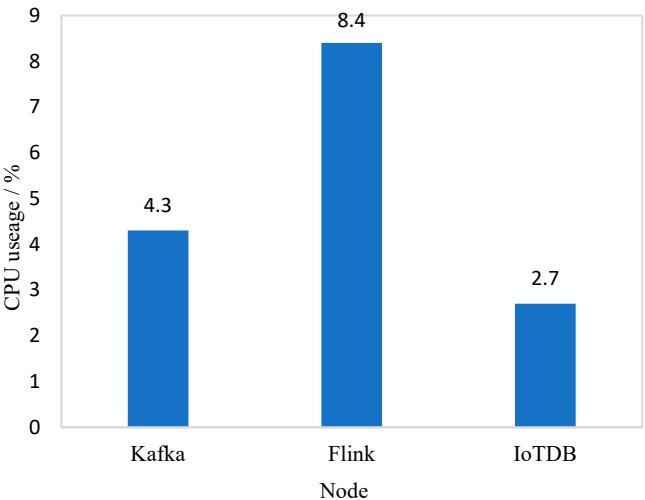

**Figure 8.** CPU usage of each node under the real dataset.

According to the experimental results, the Flink node had the highest CPU usage, reaching 8.4%. This is because data preprocessing and statistical calculation were completed by the Flink node. Kafka and IoTDB only provided data transmission and query functions, so the CPU usage was not high.

The experiment tested the memory usage of key nodes of the system, and the experimental results are shown in Figure 9. Kafka had the highest usage of memory, reaching 534 MB, while Flink and IoTDB had low usage of memory.

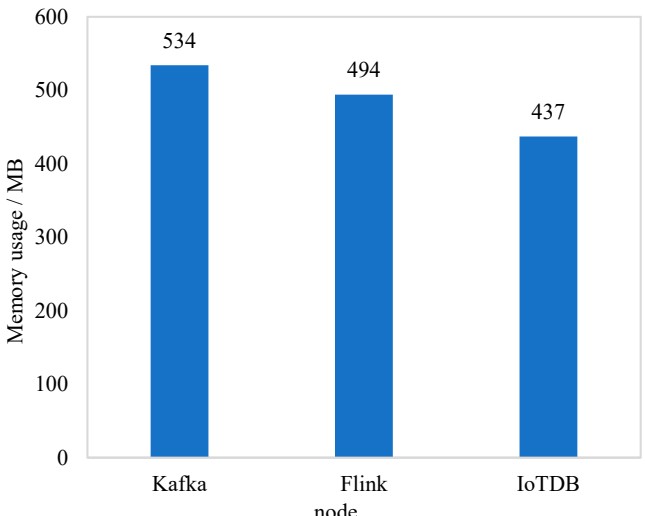

**Figure 9.** Memory usage of each node under the real dataset.

According to the overall experimental results, the resource occupation of the system was not high in the case of a real dataset simulation, and the system could effectively cope with a real hydrologic scenario on the existing server resources.

## 6. Conclusions

Hydrologic statistical analysis is significant because it helps in understanding the patterns and relationships between different hydrologic variables, such as precipitation, evaporation, runoff, and groundwater level, among others. It also enables the assessment of the reliability and validity of the data obtained from hydrologic sensors, which can be used to make informed decisions regarding water management, planning, and utilization. With the development of Internet of Things technology, sensor data has increased significantly. Hydrologic data processing has become a kind of typical big-data scene, and the traditional schemes have become difficult to use for processing and analyzing hydrologic time-series data of high concurrency in real-time. However, statistical analysis of various characteristics of hydrologic data is of great significance in establishing hydrologic models. In order to perform the real-time calculation of hydrologic stream data, HydroStreamingLib was proposed based on Flink DataStream API, and a distributed real-time statistical analysis system for hydrologic time-series data was implemented based on the algorithms available through the proposed library.

The exploitation of the proposed library resources was evaluated using real-world problems and scenarios with higher concurrency and faster flow rates. The experimental results showed that the algorithm library and the system based on it could conduct real-time and accurate analysis of hydrologic time-series data in hydrologic big-data scenarios with high concurrency and high flow rates.

Future research directions for a hydrologic statistical analysis system based on Flink could include improving the accuracy and efficiency of the analysis algorithms, incorporating more diverse data sources, and exploring new ways to visualize and interpret the results. Additionally, researchers could work on integrating the system with other hydrologic modeling tools and expanding its use to a wider range of applications, such as agriculture and meteorology. There is also room for investigation into the scalability of the system, and how it could handle increasing amounts of data. Finally, research into the security and privacy aspects of the system, particularly in the context of sharing and accessing sensitive hydrologic data, could also be a future direction.

**Author Contributions:** Conceptualization, J.S. and F.Y.; methodology and analysis, J.S.; data curation, J.S. and F.Y.; writing—original draft preparation, J.S.; writing—review and editing, J.S., F.Y. and N.N.; supervision, F.Y.; project administration, F.Y., M.Z. and D.X.; funding acquisition, F.Y. and M.Z. All authors have read and agreed to the published version of the manuscript.

**Funding:** This research was funded by the National Key R&D Program of China (2019YFE0109900); the Water Science and Technology Project of Jiangsu Province (2022003, 2022057); the Jiangsu Province Key Research and Development Program (Modern Agriculture) Project (BE2018301); and the National Natural Science Foundation of China (52179076 and 51979186).

**Data Availability Statement:** The data supporting the findings of this study are not publicly available due to privacy.

**Acknowledgments:** The authors wish to thank all members in our project team for their valuable contributions during discussions.

**Conflicts of Interest:** The authors declare no conflict of interest.

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
