# Peer review of "A Practical Yet Accurate Real-Time Statistical Analysis Library for Hydrologic Time-Series Big Data"

_water, doi:10.3390/w15040708_

Round 1

Reviewer 1 Report

Which of the many different physical processes that make up hydrology should get the most money for research?
For managing water resources and preventing floods, it's important to know how fast water can move through rivers and lakes.
What needs the most care to keep important infrastructure like dams and hydropower plants running?

Hydrologic time series analysis typically uses methods similar to the ones presented here.
What causes the inherent ambiguity of hydrologic data, as the recent study discovered?
What purpose does it serve to regularly assess our hydrologic models? What specific advantages do you gain from doing this?
How often does real-time analysis of hydrologic time series data result in major issues?
A second problem is the management of hydrologic time series data.
To maximise the potential of huge data repositories, it is necessary to understand the types of data that can be pulled from them.c
How many parts are required for Flink-based data import?
Who first told us about HydroStreamingLib, and how did they find out about it?

What will have catastrophic consequences for signature evaluation and hydrologic time series analysis?
When the preparation window executes, how does the timestamp on the data the preprocessing window is currently working with change? How regular is the action of the preprocessing window?
Which table accounts for the time period's esoteric statistical characteristics?
The Chu River's water flow is quantified using various time periods and locations, but how is this data statistically analysed?

Conclusion should state scope for future work.

The comparison of different methods using clear graphs should be explained.

Results need explanations. Additional analysis is required for each experiment to show its primary purpose.

Reviewer 2 Report

First, I would like to praise the paper and the idea of the authors. Nowadays, much of the data should be analyzed, especially in hydrology. Also, there are a lot of analysis methods with specific procedures and possibilities of the applications. 

The paper presents the same quality platform for the insight and usage of the particular method. The presentation is high-level regarding the scientific and applicable possibilities. The literature review shows all state-of-the-art of the analyzed topic.

I am proposing a minor revision. Authors should elaborate on hardware requirements for their platform, price, availability, and similar queries when some application is provided. 

Reviewer 3 Report

In the reviewed manuscript, there are several issues that need to be clarified, including:

 -        The point "Introduction" requires more information on model tests

 -        Can HydroStreamingLib also be used to analyze mteorological data?

 -        In Table 1 had been given ‘A total of different hydrologic time series analysis methods are implemented in HydroStreamingLib’ - In what way authors solved the problem about the parametric test of series homogeneity

-        Point 3.2.2 (Data Anomaly) – ‘These abnormal data often contain values that are impossible to occur  in practice. These outliers greatly affect the accuracy of subsequent statistical analysis, making anomaly removal one of the key steps for accurate and efficient data preprocessing’ – the notation is not clear, automatic deletion of outliers may result losing of valuable information. Whether outliers in time series have been analyzed, especially the reasons for the occurrence of such values?

 -        Point 5.1, line 363-364 ‘The sampling interval of this data set is 5 min, and its format is shown in Table 2 after redundant information is removed’ – I suggest adding information what % of data had been deleted.

 -        Point 5.2 – table 3, table 4, table 5, table 6 – In the tables was given the results of the statistical methods used – I suggest complementing the tables with the results of the tests indicated in Table 1

 -        Conclusions are too general - I suggest adding more detailed information
